**Data Availability Statement:** All relevant data are within the paper and its Supporting Information files.

# Effects of an industrial passive assistive exoskeleton on muscle activity, oxygen consumption and subjective responses during lifting tasks

Xishuai Qu[1], Chenxi Qu[2], Tao Ma[3], Peng Yin[1], Ning Zhao[3], Yumeng Xia[4], Shengguan Qu[1,3]*

**1** School of Mechanical and Automotive Engineering, South China University of Technology, Guangzhou, China, **2** School of Mechanical, Aerospace and Civil Engineering, The University of Manchester, Manchester, United Kingdom, **3** State Key Laboratory of Smart Manufacturing for Special Vehicles and Transmission System, Inner Mongolia First Machinery Group Co., Ltd., Baotou, China, **4** China North Advanced Technology Generalization Institute, Beijing, China

* qusg@scut.edu.cn

## Abstract

The purpose of this study was to evaluate the effects of an industrial passive assisted exoskeleton (IPAE) with simulated lifting tasks on muscle activity, oxygen consumption, perceived level of exertion, local perceived pressure, and systemic usability. Eight workers were required to complete two lifting tasks with and without the IPAE, that were single lifting tasks (repeated 5 times) and 15 min repeated lifting tasks respectively. Both of the tasks required subjects to remove a toolbox from the ground to the waist height. The test results showed that IPAE significantly reduced the muscle activity of the lumbar erector spinae, thoracic erector spinae, middle deltoid and labrum-biceps muscles; the reduction effect during the 15 min lifting task was reached 21%, 12%, 32% and 38% respectively. The exoskeleton did not cause significant differences in oxygen consumption and the perceived level of exertion, but local perceived pressure on the shoulders, thighs, wrists, and waist of the subjects could be produced. 50% of the subjects rated the usability of the equipment as acceptable. The results illustrate the good potential of the exoskeleton to reduce the muscle activity of the low back and upper arms. However, there is still a concern for the obvious contact pressure.

## Introduction

Despite the widespread use of robots instead of labors for material handling in the industrial field, many short-distance material-lifting tasks are still performed manually [1]. Compared with robots, workers' subjective initiative can represent an irreplaceable advantage in some tasks [2]. In manual handling, workers are likely to suffer from musculoskeletal disorder, of which low back pain (LBP) is the number one cause of disability in the world [3]. LBP is a serious problem that plagues industrialized countries and workers, the indirect costs caused by

**Funding:** This research was funded by the Inner Mongolia First Machinery Group Co., Ltd. State Key Laboratory of Special Vehicles and Drive Systems Intelligent Manufacturing Project Open Project (GZ2019KF001; GZ2019KF007). Visualization: T. M.;project administration: N.Z.

**Competing interests:** We confirm that the authors and this manuscript have no affiliations with or involvement in any organization or entity with any financial interest, or non-financial interest in experimental procedures. All authors include the 3 authors affiliated to Inner Mongolia First Machinery Group declare that they have no conflicts of interest to disclose, and have approved the final version of this manuscript for submission. The commercial affiliation does not alter our adherence to PLOS ONE policies on sharing data and materials.

LBP represented overall 0.68% of Spanish Gross Domestic Product [4]. From 1990 to 2016, 12.8 million individuals with LBP had increased in China [5]. Besides, LBP is among the biggest causes of absence from work [6]. Despite the increasing awareness and the abundance of research on ergonomics in the industrial field, the prevalence of musculoskeletal diseases has not decreased [7]. Common treatment options for LBP include medication, physical therapy or surgery, which can be painful, expensive and produce limited effects on recovery. Therefore, further research is needed to prevent the musculoskeletal disorder.

To reduce the incidence of musculoskeletal diseases of workers during manual material handling (MMH) work, off-body mechanical aids, such as trolleys and forklifts, are used to carry toolbox beyond human capability. Although it can effectively reduce the situation of workers carrying heavy loads [8], lifting aids are often not used due to their constraints [9]. On-body assistive devices, such as widely-sold back belts, have no definitive research evidence to show that they are effective in protecting or preventing injury when workers are involved in MMH tasks [10–12]. In recent years, people have paid more and more attention to wearable robot technology (including exoskeletons) to help workers perform manual lifting tasks without risk [13]. Exoskeleton is a new type of wearable assistive device that can reduce the risk of musculoskeletal disorder by aiding the human body.

Exoskeletons are generally classified as active and passive. The active types usually use the drive system (motor, hydraulic system or pneumatic system, etc.) to enhance human strength and reduce the body's energy consumption. Naruse et al. [14] developed an electric motor-assisted device to assist trunk flexion and extension to reduce the load on the waist; however, the weight of the second-generation prototype is still 6.5 kg, which is too heavy for workers in a bent position. The smart suit developed by Takayuki et al. could reduce about 14% of muscle fatigue in the bending process [15]. Their device was driven by a 24V DC motor, and it was hard to incorporate into the workplace because the motor was difficult to carry around. Besides, several other well-known active exoskeletons, such as HAL, Muscle Suit and BLEEX, were large and expensive so that they were not suitable for workers [16–18]. At present, the price, stability and versatility of active exoskeletons have not been recognized by the industry, and some active exoskeletons dedicated to industrial applications are still in the laboratory [19, 20].

Passive exoskeletons use elastic members to store and release energy during lifting works. Some passive exoskeletons have entered the marketing stage. Several passive exoskeletons were shown to reduce the muscle activity of the lower back significantly, such as Happyback, Personal Lifting Assist Device (PLAD), Laevo and Bendezy. Happyback is composed of fiberglass rods with a chest harness, waist belt and leg units attached to it [21]. PLAD is made up of elastic elements, which support part of the weight of the upper body when bending down [22]. Laevo is a chest and back supporting exoskeleton composed of flexible tubes that transfer part of the load to the chest and legs [23]. Bendezy consists of a back unit and straps that wrap around the shoulders, back and legs; springs bear part of the weight. Most of these passive exoskeleton studies only focused on the protection of the lower back muscles but ignored the fatigue of the arm muscles in the lifting task. There was little research on the local discomfort caused by the exoskeleton. Moreover, experiments for exoskeleton usually emphasized the collection of EMG, lacked the detection of other physiological indicators. The test results could not fully reflect whether an exoskeleton met the needs of subjects.

In response to the above problems, an novel industrial passive assistive exoskeleton (IPAE) was developed by us to reduce the risk of disorder to both low back muscles and arm muscles of workers during lifting works. The IPAE's total weight is only 4 kg because of the structural optimization and the surface is wrapped by flexible fabrics, which improves the comfort of wearing. Some early subjects have felt the differences in whether they wore the IPAE working in the stooped posture; however, the objective effectiveness of this exoskeleton to relieve

fatigue was unclear, and there is still a lack of quantitative indicator to evaluate the subjective feelings. Therefore, in this study the effects of exoskeletons on the wearer's muscle activity, oxygen consumption, local perceived pressure, perceived fatigue level and systemic usability were investigated by testing each subject finishing two types of simulated lifting tasks to evaluate the IPAE fully.

## Materials and methods

### Passive exoskeleton

IPAE is a passive wearable exoskeleton that consists of a back support, waist elastic units and leg supports connected in sequence. The structure diagram of IPAE used in the tests is shown in Fig 1. The IPAE was worn by subjects like a backpack. After putting it on, the straps on the chest, waist, thighs and wrists needed to be fixed and adjusted. When the upper body is lowered, energy is stored in the waist elastic elements; on the ensuring upward phase, the stored energy is released, thereby reducing the activity of the lower back muscles. The lifting object is fixed with the hooks and the straps connecting shoulders and wrists transfer part of the weight of the object to the shoulders during lifting tasks to relieve the fatigue of the arm muscles. When subjects restoring upright, part of the box weight is transferred to the shoulders and back support by the straps, and the elastic unit releases the potential energy to provide assistance. After returning to an upright position, the box weight is partly transferred to the shoulders. A researcher was required to assist in the initial wearing, and the entire wearing process takes about 2 min. Before the formal tests, the subjects needed to finish normal walking and lifting actions until the straps and elastic elements were adjusted to the appropriate range.

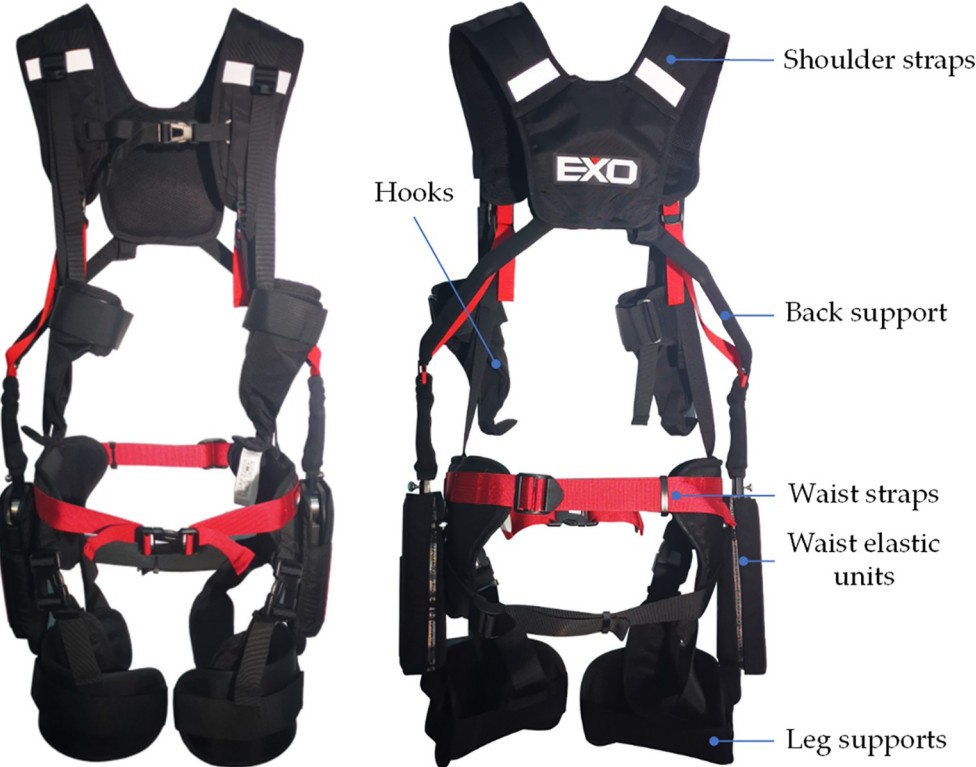

**Fig 1. The structure diagram of IPAE used in the tests.** It consists of a back support, waist elastic units, leg supports, hooks, shoulder straps and waist straps showed in two views: (a) front view; (b) rearview.

## Participants

Eight healthy adult male workers (right inertial hand) with no history of muscle injury and LBP in the past three months were invited to participate in this study. The subjects read and signed an information and consent form approved by the South China University of Technology Research Ethics Board. To reduce the influence of physical factors, the selected subjects were 27.4±4.1 years old, 73.2±8.1 kg in weight and 174±5.4 cm in height. All subjects read the test process and precautions in detail before the tests and signed the consent form. The subjects did not engage in vigorous exercise one week before the tests, and a 5 min warm-up was performed before the tests.

## Instrumentation

Previous research showed that there was no significant difference between the left and right EMG signals [24]. Four-channel portable Flexvolt Bluetooth EMG Sensor with a sampling rate of 2048 Hz was used to collect surface EMG data of the right four muscles: thoracic (T9) erector spinae (TES), lumbar (L4) erector spinae (LES), middle deltoid (MD) and labrum-biceps (LB) respectively. The position of the electrodes was shown in Fig 2. A pair of Ag/AgCl electrodes parallel to the orientation of the muscle fibers (distance between electrodes: 2 cm) was placed over each muscle belly. The reference electrode placed on the electrically neutral side of the vertical muscle fiber orientation. Before fixing the electrodes with medical tape, the skin surface was shaved and cleaned with alcohol, and finally sprayed by antiperspirant to prevent electrode displacement and signal loss due to sweating. Oxygen consumption was collected with VO2 Master Health Sensor equipment. The subjects had worn the equipment for information entry and calibration before formal measurement. Each subject finished 3 maximum voluntary contractions (MVCs) intervals separated by 1 min before the start of all tests. After completing the maximum back extensor force (MVE) tests on the subject given by Christy A. Lotz et al. [25], a toolbox with a weight of 20% MVE was set as the load to be lifted.

## Testing procedures

**Preparation.** The laboratory used for sessions maintains a constant temperature of 22˚C, and all irrelevant electronic equipment had been cleaned to reduce signal interference before the tests. After subjects entered the laboratory, the staff demonstrated how to wear and use test-related equipment and introduced the test process in detail until the subjects were proficient to complete the lifting task at a roughly uniform speed with the metronome. After

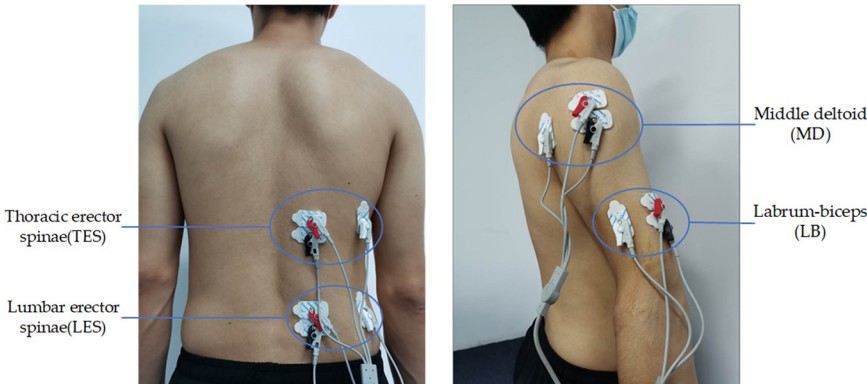

**Fig 2.** The electrodes placed on (a) the low back and (b) the upper arm. The EMG of thoracic (T9) erector spinae (TES), lumbar (L4) erector spinae (LES), middle deltoid (MD) and labrum-biceps (LB) are collected by Ag/AgCl electrodes.

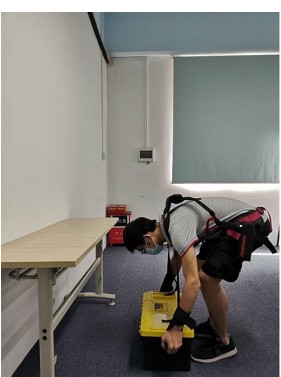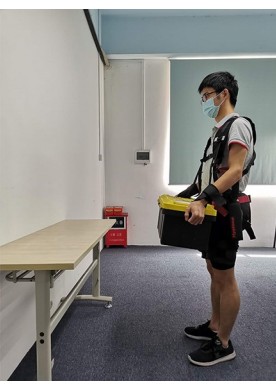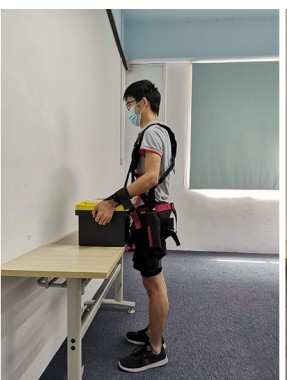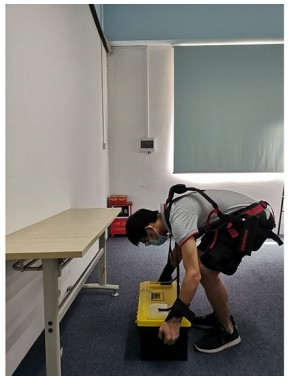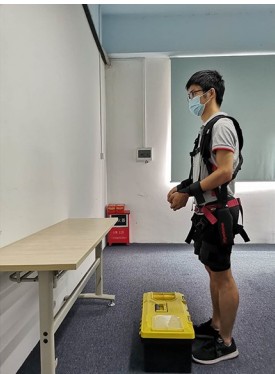

**Fig 3. A subject performing the whole lifting process with the exoskeleton.** The toolbox was set at 20% MVE of the subject and the table had been adjusted to the waist height.

training, the subjects performed the single lifting task for total 5 times with an interval of 1 min each time. Then the subjects finished two 15 min lifting tasks (with or without exoskeleton) in random order, and the interval between two tasks was more than 12h. It is necessary to confirm that the subjects did not experience muscle discomfort before all tests.

**Lifting tasks.**   Each lifting task started when subjects remained relaxed and upright. The lifting process is shown in Fig 3. The tasks were executed in the sagittal plane, using free postures. Subjects lifted the toolbox on the ground in front of them. After returning to the upright position, the toolbox was placed on a table at the same height as the waist. Then the subjects put the toolbox back on the ground and returned to the starting posture. The entire process was controlled by a metronome to maintain about 12s after training. Each subject completed 75 groups in 15 min lifting task. The subjects' oxygen consumption data and the EMG data of four muscles tested were collected throughout.

**Subjective responses.**   After the single lifting task, perceived musculoskeletal pressure was rated by Local Perceived Pressure (LPP) method adapted from Van der Grinten [26]. LPP ratings ranging from 0 (no pressure at all) to 10 (extremely strong pressure) were used to assess the musculoskeletal pressure of areas that are in close contact with IPAE, including the waist, shoulders, wrists, and thighs. During the 15 min lifting tasks, subjects rated the current fatigue level referring to Borg's Rate of Perceived Exertion Scale (Borg RPE 6–20) every minute. After the task, the subjects needed to use the System Usability Scale (SUS) evaluating of IPAE [27]. The SUS consists of ten questions rated from one (strongly disagree) to five (strongly agree). The score of 0–100 can reflect the degree of acceptance of the exoskeleton by subjects. The score over 70 is generally considered acceptable for this exoskeleton.

## Data processing

All original EMG signals were rectified and then a second-order Butterworth filter was used for 20–500 Hz bandpass filtering. Finally, 30 Hz and 50 Hz notch filter were used to eliminate ECG contamination and eliminate power frequency interference in the signal [28]. The RMS of EMG data was calculated to determine the signal amplitude, and it was normalized according to the maximum EMG obtained during the MVC test to compare different subjects (MVC % = RMS/ RMSmax*100%).

## Statistical analysis

The independent variables in the tests were whether to wear IPAE and the test time (i.e. 15 min). The dependent variables were the RMS amplitude of four muscles, oxygen consumption

**Table 1. The result of EMG tests.**

| Muscle | Condition | Single/15 min | RMS amplitude (MVC%) | Standard deviation (MVC%) | P value |
|--------|-----------|---------------|----------------------|---------------------------|---------|
| LES | I | single | 25.5 | 5.6 | 0.002 |
| LES | NI | single | 34.7 | 5.1 | |
| LES | I | 15 min | 29.5 | 6.9 | 0.041 |
| LES | NI | 15 min | 37.4 | 6.8 | |
| TES | I | single | 17.6 | 3.0 | 0.107 |
| TES | NI | single | 20.8 | 2.9 | |
| TES | I | 15 min | 19.1 | 2.6 | 0.027 |
| TES | NI | 15 min | 21.6 | 2.9 | |
| MD | I | single | 13.6 | 2.8 | 0.009 |
| MD | NI | single | 17.8 | 3.4 | |
| MD | I | 15 min | 17.7 | 4.9 | 0.017 |
| MD | NI | 15 min | 26.1 | 4.4 | |
| LB | I | single | 13.2 | 1.7 | 0.001 |
| LB | NI | single | 17.6 | 1.7 | |
| LB | I | 15 min | 14.7 | 1.2 | 0.000 |
| LB | NI | 15 min | 23.7 | 1.3 | |

The summary of means and standard deviations of RMS amplitude for all muscles (LES, TES, MD and LB) and tasks (Single lifting task and 15 min lifting task) was listed. P<0.05 means the test result is significant.

and Borg RPE scale. The paired sample t-test was used to evaluate the differences in RMS and oxygen consumption with and without IPAE. After the K-S test, the Borg scale did not violate the assumption of normality, and repeated measures analysis of variance (ANOVA) was used. All statistical analyses were performed by SPSS for Windows. The significance was set at p<0.05.

## Results

### Muscle activity

LES, MD and LB muscle activity was significantly lower (p<0.05) with the IPAE, but not for the TES in the single lifting task (Table 1). Fig 4 shows the RMS amplitude of the two tasks. In the single lifting task, IPAE helped the LES muscles the most, reducing muscle activity by 26%. During the 15 min lifting task, MD and LB muscle activities were reduced by 32% and 38%, the IPAE's effect on the upper arm muscles was more obvious.

### Oxygen consumption

Fig 5 shows the average relative oxygen consumption of 8 subjects with and without IPAE during the 15 min tests. The paired sample t-test result showed that there was no significant difference in relative oxygen consumption between the two conditions (p = 0.59). The average value of all subjects over the entire 15 min was 16.02±1.64 ml/kg/min (IPAE) and 15.98±1.55 ml/kg/min (NO-IPAE).

### Subjective responses

**Borg's Rate of Perceived Exertion Scale.** The Borg RPE scale under the two conditions is shown in Fig 6. After repeated measure ANOVAs test, the main effect of time was significant for both conditions (p = 0.047<0.05), that is, the scale would increase significantly with time.

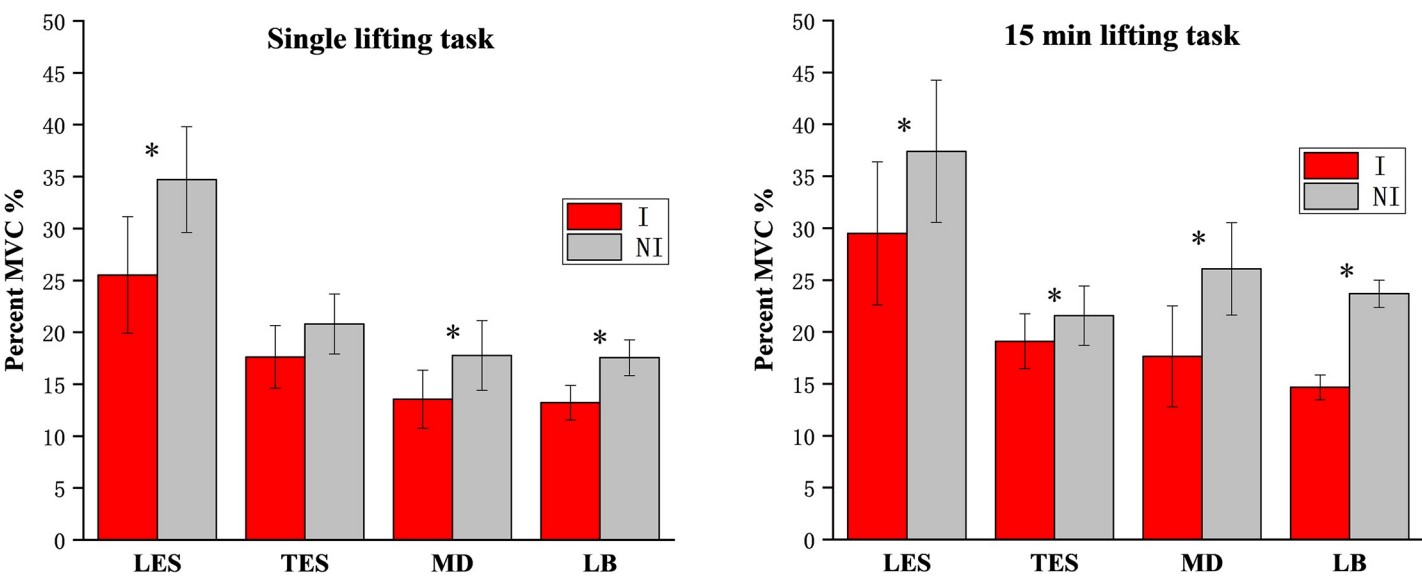

**Fig 4. Mean RMS amplitude.** The values include lumbar erector spinae (LES), thoracic erector spinae (TES), middle deltoid (DM) and labrum-biceps (LB) with (I) and without (NI) IPAE. (**a**)single lifting task; (**b**)15 min lifting task. Significant results (p < 0.05) are marked with an *.

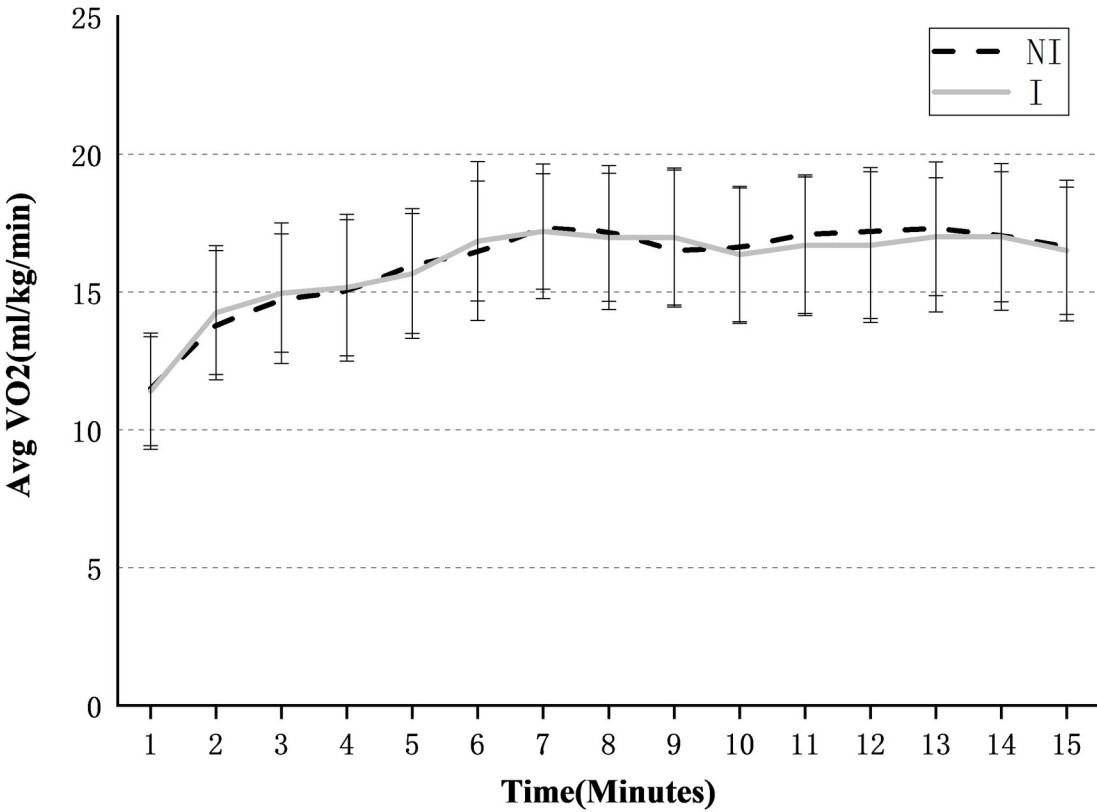

**Fig 5. Average relative oxygen consumption.** The oxygen consumption of 8 subjects was checked over time under the IPAE(I) and No-IPAE(NI) conditions.

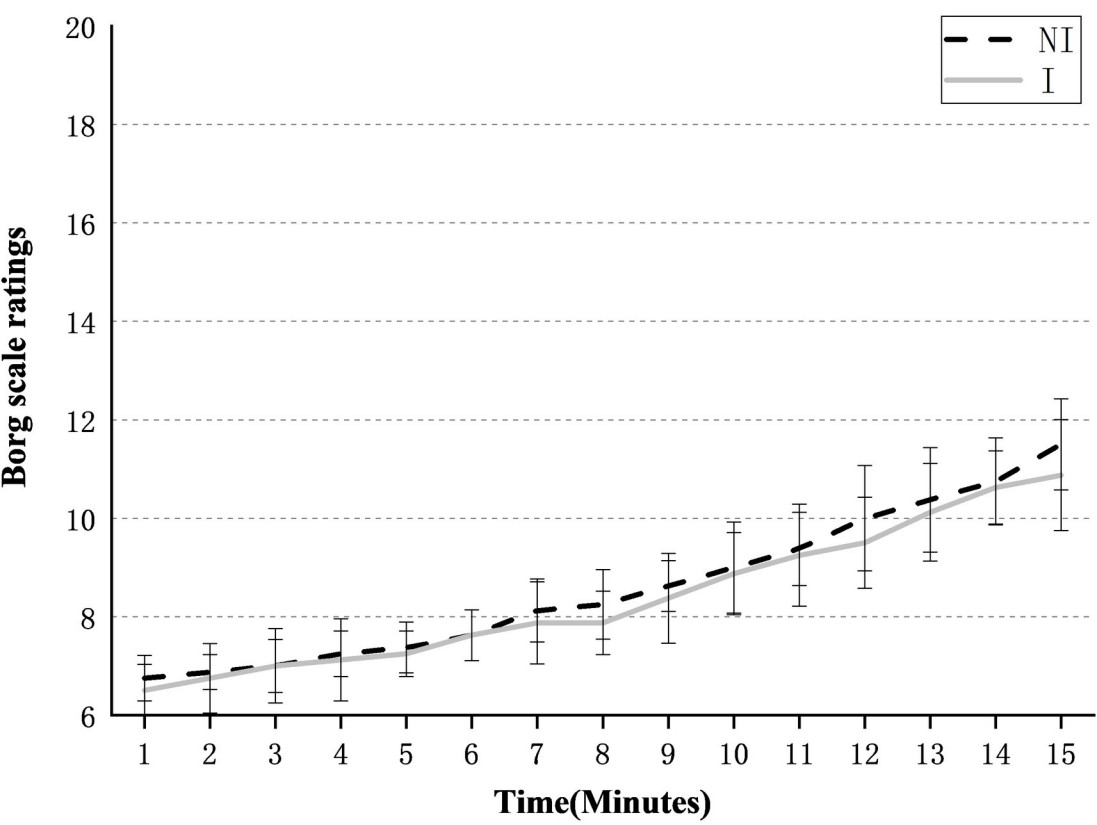

**Fig 6. Mean Borg scale ratings of 8 subjects over time for the IPAE(I) and No-IPAE(NI) conditions.**

There was no time and equipment interaction to indicate that IPAE affects the Borg scale, and whether to wear IPAE did not cause a significant difference. At the end of the tests, the Borg scale was 10.88±1.13 with IPAE, which was lower than 11.5 (±0.93) without IPAE.

**Local perceived pressure.** The mean scores of the local perceived pressure of all subjects are shown in Fig 7. The scores of the four parts most in contact with IPAE are: Shoulders (2.38) > Thighs (1.75) > Wrists (1.38) > Waist (1). Subjects felt the greatest contact pressure on the shoulders, followed by the thighs, and less pressure on the wrists and waist.

**Usability.** Fig 8 shows the system usability scores of 8 subjects. All subjects rated the system usability scores, and four subjects rated SUS scores above 70 points, which was higher than the acceptable usability standard, and other 4 subjects' scores were rated within the accepted critical value range.

## Discussion

The test results of EMG show that whether the single lifting tasks or the 15 min lifting tasks, IPAE can reduce the muscle activities of the low back and upper arms. IPAE provided greater assistive effects on upper arm muscles compared to low back muscles during the 15 min tests. This is consistent with the feedback from the subjects. Wearing IPAE would make arms feel much easier to lift the toolbox. Due to the decrease in muscle activity with IPAE, it can be expected that when workers wear IPAE during bending over and lifting, muscle fatigue of the low back and upper arms will be reduced. Also, the studies from Granata et al. [29] have shown that when the spine is overloaded, waist injury would occur. IPAE reduces muscle activity in the low back and also helps protect the spinal structure.

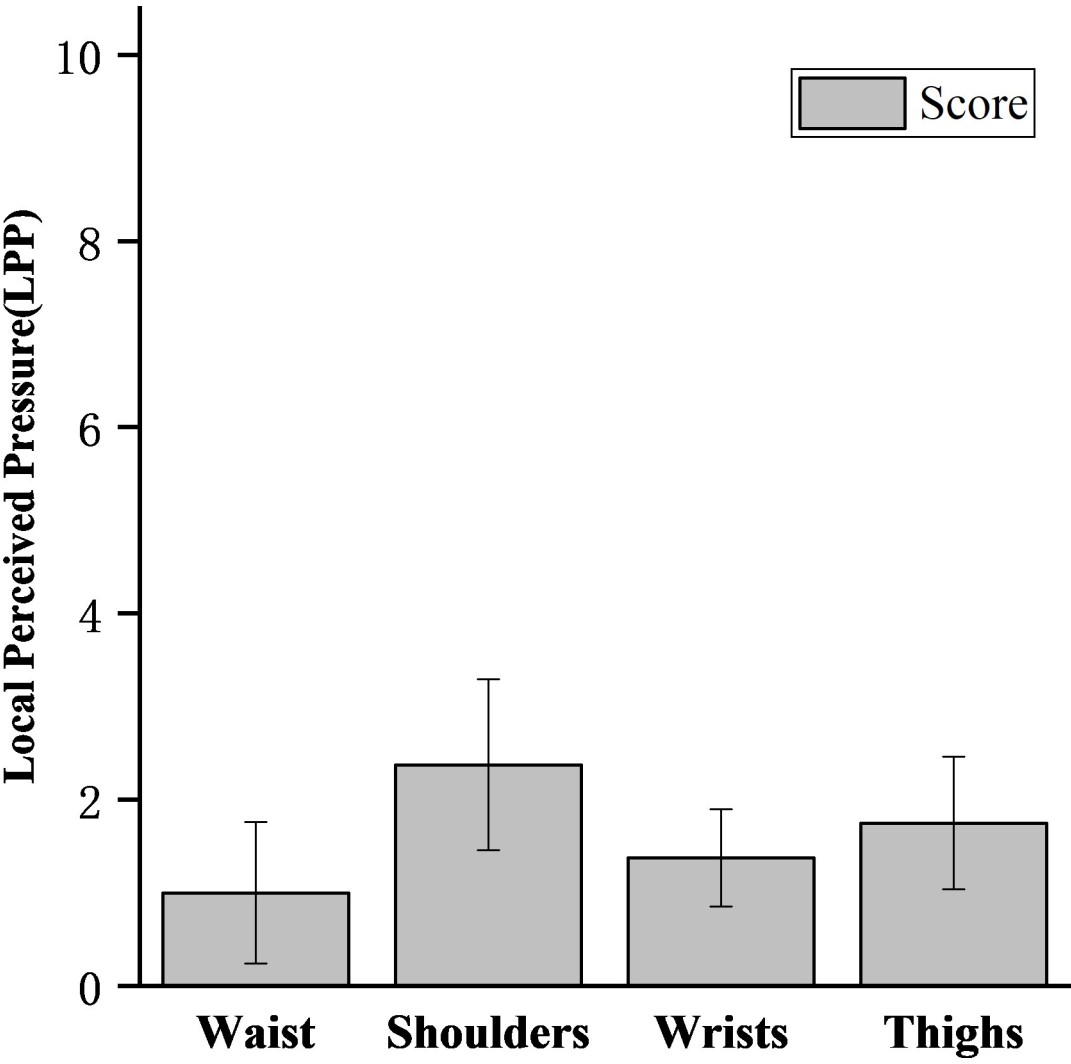

**Fig 7. Mean local perceived pressure of 8 subjects for the single lifting task with IPAE.**

With/without IPAE does not affect subjects' oxygen consumption, which meets the expected assumption and is also consistent with the conclusion drawn by Brett H. Whitfield et al. [30]. Wearing a passive exoskeleton does not significantly increase or reduce the oxygen consumption index in repeated lifting tasks. The average relative oxygen consumption without IPAE was 15.98 ml/kg/min, which is lower than the 17.8 ml/kg/min measured by Brett H. Whitfield et al. This difference may be due to the different lifting task, subject aerobic capacity, free choice of lifting postures, and lifting weights. The Borg RPE scale increased significantly over time during the 15 min tasks, but there was no significant difference with or without IPAE.

Combining the results of oxygen consumption and Borg RPE, IPAE does not significantly reduce the fatigue of subjects. Even if it can effectively reduce the muscle activity of the low back and upper arms, IPAE did not reduce the body's energy requirements and oxygen consumption during lifting tasks. But the studies by Baltrusch et al. has showed a reduction in energy consumption during lifting with the LAEVO [31]. When wearing IPAE for handling work, it will result in a significant increase in some other muscle activities, such as shoulders

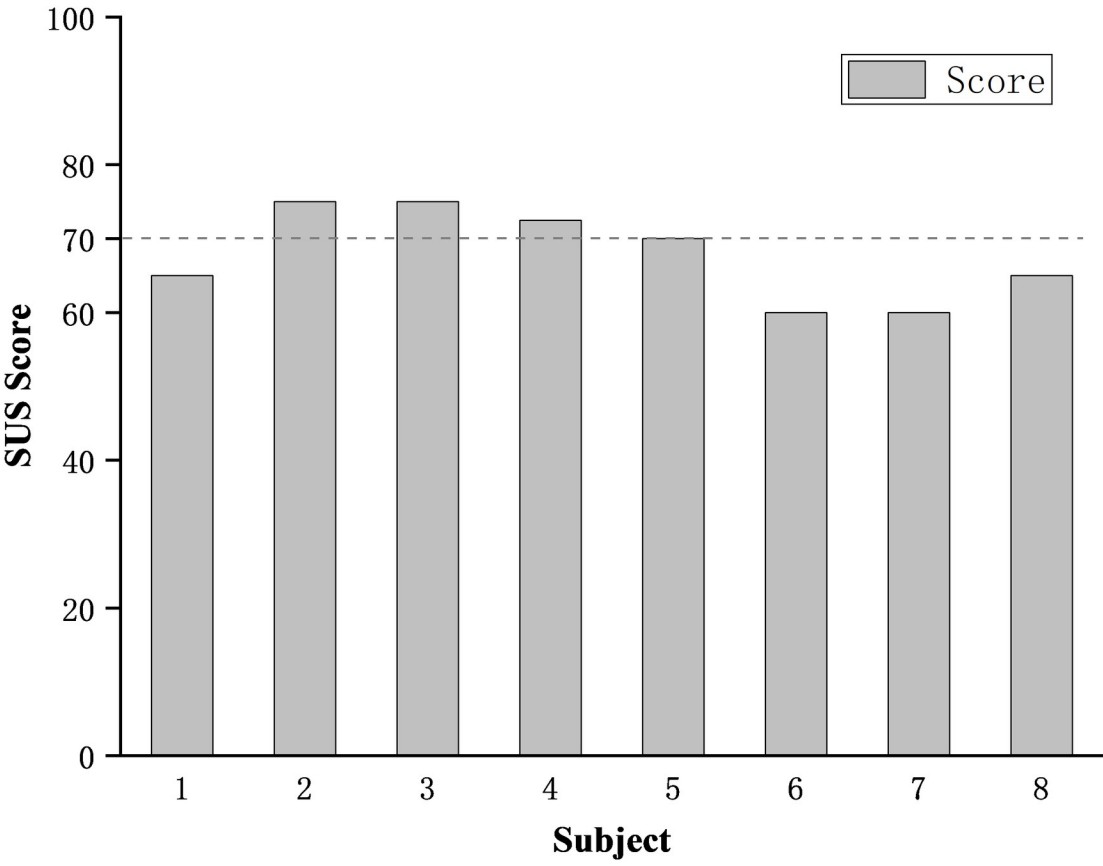

**Fig 8. Participant SUS ratings of the exoskeleton.**

and thighs. The LPP scores of subjects in Fig 6 indirectly verified this. Therefore, for workers equipped with IPAE, they should not increase the workload and the working frequency or extend the working hours.

The LPP score of shoulders with the highest perceived pressure was 2.38±0.92, indicating that the additional pain or injury will not be produced by wearing the IPAE to the worker. But it should be noted that the LPP score was collected after the subjects had finished single lifting tasks. As lifting time increases, the LPP score may increase correspondingly. The significant contact pressure was felt on the thighs, and the exoskeleton designed by Nilson et al. [32] has a similar phenomenon, which may be due to the tight fixation of the leg units. Wrist and shoulder straps are designed to reduce the burden on the arm muscles of workers when subjects carry out handling tasks. But this structure causes significant contact pressure on the shoulders and wrists. The narrow shoulder straps may be one of the reasons for the highest LPP score. Dispersing pressure over a large area is a common method to reduce the discomfort of exoskeleton design [33]. Adding soft pads to the exoskeleton can also prevent skin injuries [34]. Hence, the improvement plan of the IPAE ought to include measures to widen the straps and add soft pads. All the subjects did not report that there was obvious stress caused by the exoskeleton in other structures of the body.

50% of the subjects rated the IPAE as having acceptable usability. Subjects with a usability score of less than 70 were generally neutral about whether they would like to use the exoskeleton frequently. It can be speculated that longer use may reduce the acceptance of the exoskeleton. They also reflected that the appearance of the IPAE looked moderately awkward. In the

future, it is necessary to optimize the appearance and weight loss of IPAE to improve the acceptance of the subjects. The SUS scores might be negatively affected by additional testing equipment, repeated task training, and assisted wearing. In actual industrial applications, these negative effects will be greatly reduced. This study also finds that the score of SUS may also be affected by the age of the subjects in this work. The SUS scores of 4 older subjects (30.5±3.1 years, 63.8±4.8 scores) were significantly lower than other 4 younger subjects (24.3±1.7 years, 71.9±4.7 scores). The younger subjects generally thought that IPAE was easy to learn, and they had a negative attitude to learning many things before using the IPAE from the specific scores of SUS.

It is worth noting that the small sample size is a limitation of the study. Due to the limitation of the EMG channels, the signal acquisitions of subjects' thighs, abdominal and forearm muscles were not completed at the same time. In the future, an 8-channel device will be used to complete the tests. To prevent musculoskeletal injuries to the subjects, the experiment only set 15 min to simulate the industrial repeated lifting tasks. After optimizing the design of IPAE, it will be tested for a longer time and completed in the actual factory. More workers will be invited to wear IPAE in actual lifting tasks and give LPP and SUS scores, which will help to evaluate the PLAD more accurately. Besides, the adjustable range of straps was too large in order to meet subjects of different sizes, and the subjects often felt the straps were too loose or too tight. Therefore, IPAE of different sizes will be considered to adapt to different populations.

## Conclusions

The test showed that IPAE significantly reduced low back and upper arm muscle activity for both finishing the single lifting task and the 15 min repetitive lifting task. When finishing the intermittent number of lifting tasks, IPAE had the most significant effect on the lumbar erector spinae by reducing the muscle activity of 26.5%. The muscle activity of deltoid and labrum-biceps muscles was more obviously reduced (32.3%, 38.1%) during the long-time repeated lifting tasks. Whether to wear IPAE would not cause significant differences in the subjects' oxygen consumption and perceived fatigue level during lifting tasks. The LPP scores indicated that IPAE would additionally increase the perceived pressure on the shoulders, wrists, and thighs. 50% of the subjects' feedback noted that the IPAE was acceptable. In summary, the IPAE significantly reduced the muscle fatigue of both the low back and upper arms of subjects during lifting works, but the test results reveals low effects and high discomfort at the same time as a passive exoskeleton. The result of the oxygen consumption test shows that it is unreasonable to require workers wearing IPAE to improve work efficiency or extend working hours. The discomfort caused by IPAE on the part of the subject's contact area reduced the user's level of satisfaction, which may be a common problem with passive exoskeletons. Future design improvements to IPAE shall focus on solving this discovered problem.

## Supporting information

**S1 Dataset.**
(XLSX)

## Acknowledgments

The authors would like to thank the company Hyetone for supporting us. Their workers were willing to assist us in the tests as subjects and provided many suggestions for the improvement of the exoskeleton. Special thanks are also extended to Zhiyuan Lu, Dezheng Zeng, Yupeng

Gao, Aimin Xu, Haidi Qin, Yuqing Cai, Yalong Zhang, Fuqiang Lai, Xiongfeng Hu for their suggestions to this research study.

## Author Contributions

**Conceptualization:** Shengguan Qu.

**Funding acquisition:** Yumeng Xia.

**Investigation:** Xishuai Qu.

**Methodology:** Shengguan Qu.

**Project administration:** Ning Zhao.

**Software:** Chenxi Qu.

**Visualization:** Tao Ma.

**Writing – original draft:** Xishuai Qu.

**Writing – review & editing:** Peng Yin.

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
