## [Decision Letter · Decision Letter 0]

1 Oct 2020

PONE-D-20-26515

Effects of an industrial passive assistive exoskeleton on muscle activity, oxygen consumption and subjective responses during lifting tasks

PLOS ONE

Dear Dr. Qu,

Thank you for submitting your manuscript to PLOS ONE. After careful consideration, we feel that it has merit but does not fully meet PLOS ONE’s publication criteria as it currently stands. Therefore, we invite you to submit a revised version of the manuscript that addresses the points raised during the review process.

For acceptance, it is crucial that you adress the following points:

-It must be included in the discussion that the small sample size is a limitation of the study.

-A conflict of interest must be declared by the 3 authors affiliated to Inner Mongolia First Machinery Group, which manufactured and funded this research.

-In order to meet PLOS ONE criteria for papers that describe new tool, methods or software for applications, the report must meet the criteria of utility, validation, and availability, which are described in detail at                          http://journals.plos.org/plosone/s/submission-guidelines#loc-methods-software-databases-and-tools. Especially, you should explain why the new tool is an improvement over existing options in some way.

-You should carefully respect the helpful comments of reviewer 2 when preparing a revised version of the manuscript. Especially, you must adjust your conclusions, which are currently not fully supported by the results. You must mention that the study reveals low effects and high discomfort at the same time.

We look forward to receiving your revised manuscript.

Kind regards,

Peter Schwenkreis

Academic Editor

PLOS ONE

Journal Requirements:

2.   Please declare in the competing interests that three of the authors are affiliated to Inner Mongolia First Machinery Group.

1) Within your Competing Interests Statement, please confirm that this commercial affiliation does not alter your adherence to all PLOS ONE policies on sharing data and materials by including the following statement: "This does not alter our adherence to PLOS ONE policies on sharing data and materials.” (as detailed online in our guide for authors http://www.PLOSone.org/static/editorial.action#competing).

ii) If this adherence statement is not accurate and there are restrictions on sharing of data and/or materials, please state these. Please note that we cannot proceed with consideration of your article until this information has been declared.

2)     Could you please clarify whether the validated device is being marketed/commercialised? If so, please state the name of the company manufacturing/marketing this device as well as the name under which the device is marketed.

3)    Please clarify whether the validated device has any attached patents and, if so, please state who is the owner of such patent. Declare this as a competing interest if any of the authors is the owner of the device patent or if the patent owner contributed in any way to the present submission (e.g, by providing funding, materials etc).

4. Please include additional information regarding the questionnaire or data collection tool used in the study and ensure that you have provided sufficient details that others could replicate the analyses. For instance, if you developed a questionnaire as part of this study and it is not under a copyright more restrictive than CC-BY, please include a copy, in both the original language and English, as Supporting Information.

Reviewers' comments:

Reviewer's Responses to Questions

**Comments to the Author**

1. Is the manuscript technically sound, and do the data support the conclusions?

Reviewer #1: Partly

Reviewer #2: Partly

2. Has the statistical analysis been performed appropriately and rigorously? 

Reviewer #1: Yes

Reviewer #2: Yes

3. Have the authors made all data underlying the findings in their manuscript fully available?

Reviewer #1: Yes

Reviewer #2: No

4. Is the manuscript presented in an intelligible fashion and written in standard English?

Reviewer #1: Yes

Reviewer #2: Yes

5. Review Comments to the Author

Reviewer #1: Interesting study unfortunately with a small sample size and reasonable time of workload. The results illustrate a good potential of the passive exoskeleton to reduce muscle activity of the low back and upper arms, but users perceived more pressure on the shoulders, thighs, wrists and waist. Only 50 % of the users rated the usability of the equipment as acceptable. So the questions is whether acceptance is farther reduced by wearing the exoskeleton for a longer time?

Validity of the studie respectively to the benefit for the users might be better by a bigger sample size.

Reviewer #2: The authors present an evaluation study for a developed passive exoskeleton. The following remarks might be useful to consider when finalizing the paper.

Introduction section

- lower back pain should be linked to a musculoskeletal disorder rather than injury

- LPP is named whithout introdcution of the meaning

- sources are quite old (e.g. LBP data from 2008) and should be updated

- there are studies published with subjective evaluation of exoskeletons (e.g. from Graham et al. 2009 and Dewi et al. 2018). This should be corrected.

Material and Methods section

- it is not clear, whether and how the upper arms are supported by the exoskeleton

- the EMG system seem to be wired - have there been any contacts with the exoskeleton? - this might have caused interferences

Testing procedures section

- why did the subjects rate LPP only after the single lifting tasks?

- it should be explained why the single lifting tasks were performed in adition to the repetitive lifting tasks

- Figure 3 shows very unergonomic lifting, why was this way of lifting chosen?

Results section

- only half of the users rate the system usable...

- there are only effects for muscle activity not for perceived exertion or oxygen cosumption - what is a possbile explanation for this?

Discussion Section

- Reducing muscular activity does not reduce risk of injury per se

- It should be discussed what possible long term effects could be

- It is stated that age could have an effect on the SUS scores - with 27 years average and 4 years SD?

- there are studies showing a reduction in energy consumption (e.g. Baltrusch et al. 2018 with LAEVO), this should be taken into account

- the authors do not discuss that other structures of the body might have a higher load due to the use of the exoskeleton

Conclusion Section

- The study reveals low effects and high discomfort at the same time. This should be mentioned accordingly.

- In total the results currently do not support fully the drawn conclusions (e.g. reduction of muscular injury).

PLOS Data policy:

The authors state that all data are fully available without restriction - yet there is no link to a respository given in the documents.

6. PLOS authors have the option to publish the peer review history of their article (what does this mean?). If published, this will include your full peer review and any attached files.

Reviewer #1: No

Reviewer #2: No

---

## [Author Response · Author response to Decision Letter 0]

9 Dec 2020

Response to Reviewers

Dear Editor Peter Schwenkreis and Reviewers:

We would like to thank you for reviewing our manuscript (Effects of an industrial passive assistive exoskeleton on muscle activity, oxygen consumption and subjective responses during lifting tasks) and giving us many valuable and constructive comments. We have carefully modified our manuscript based on your comments, which were highlighted in yellow background in the revised manuscript.

Response to Academic Editor:

Comment 1:

Response to Comment 1:

Dear editor, thank you for the PLOS ONE style templates you provided. We have checked the manuscript repeatedly according to the format requirements in the template, to meet all the format requirements as much as possible. Modifications include: file renaming, modification of the article title format on the homepage, deletion of the postcode from Affiliations, listing corresponding author’s initials in parentheses after the email address and using Fig instead of Figure. The "Vancouver" style is adopted for the citation format in the main text and all figures have appeared after the first quoted paragraph. We have uploaded our figure files to the Preflight Analysis and Conversion Engine (PACE) digital diagnostic tool. All the figure files has been checked and changed in the manuscript.

Comment 2:

Please declare in the competing interests that three of the authors are affiliated to Inner Mongolia First Machinery Group.

1) Within your Competing Interests Statement, please confirm that this commercial affiliation does not alter your adherence to all PLOS ONE policies on sharing data and materials by including the following statement: "This does not alter our adherence to PLOS ONE policies on sharing data and materials.” (as detailed online in our guide for authors http://www.PLOSone.org/static/editorial.action#competing).

ii) If this adherence statement is not accurate and there are restrictions on sharing of data and/or materials, please state these. Please note that we cannot proceed with consideration of your article until this information has been declared.

2) Could you please clarify whether the validated device is being marketed/commercialised? If so, please state the name of the company manufacturing/marketing this device as well as the name under which the device is marketed.

3) Please clarify whether the validated device has any attached patents and, if so, please state who is the owner of such patent. Declare this as a competing interest if any of the authors is the owner of the device patent or if the patent owner contributed in any way to the present submission (e.g, by providing funding, materials etc).

Response to Comment 2:

 Thanks for pointing this out. The Competing Interests Statement is as follows:

 We confirm that the authors and this manuscript have no affiliations with or involvement in any organization or entity with any financial interest, or non-financial interest in experimental procedures. All authors include the 3 authors affiliated to Inner Mongolia First Machinery Group declare that they have no conflicts of interest to disclose, and have approved the final version of this manuscript for submission. The commercial affiliation does not alter our adherence to PLOS ONE policies on sharing data and materials. 

 At present, we have reached a commercial cooperation with Guangzhou Heytone Company, and have tried commercial promotion and test marketing of IPAE. The marketing name is ‘STRONG HANDS’. Now the validated device has not attached patents.

Comment 3:

We suggest you thoroughly copyedit your manuscript for language usage, spelling, and grammar. If you do not know anyone who can help you do this, you may wish to consider employing a professional scientific editing service. 

Response to Comment 3:

 We have checked the manuscript for language usage, spelling and grammar thoroughly. Five doctors with extensive experience in writing English manuscripts were invited to read and check the manuscript. All relevant language changes were highlighted in the text. At the same time, thanks for your recommendation on language editing services. If there is any need in the future, we will give priority to the cooperative websites you recommend.

Comment 4:

Please include additional information regarding the questionnaire or data collection tool used in the study and ensure that you have provided sufficient details that others could replicate the analyses. For instance, if you developed a questionnaire as part of this study and it is not under a copyright more restrictive than CC-BY, please include a copy, in both the original language and English, as Supporting Information.

Response to Comment 4:

 Dear reviewer, the questionnaires used throughout the experiment mainly include the Borg RPE, the Local Perceived Pressure(LPP) and the System Usability Scale(SUS). Three scales had been appropriately modified to better meet the needs of experimental testing. The Borg table is shown in Table 1 below, the different colors help the subjects to distinguish different grades better. The local perception pressure meter is shown in Table 2, and the system availability scale is shown in Table 3 below. 

Table 1. Borg RPE(6-20)

6 Rest

7 Not tired at all

8 Feel a little tired or not tired, your breathing is gentle

9 

10 Feel slightly tired, breath slightly rising but still steady

11 

12 Feel slightly tired, breath faster than 5

13 

14 Moderately strong-feeling tired, short of breath

15 

16 Very strong-the sensation that occurs during very strenuous exercise, feeling extremely tired

17 

18 Super strong-this is the feeling that occurs under extreme strenuous exercise, extreme exhaustion, not last until the end of the exercise, your breathing is very laborious, and you cannot talk to people.

19 Extremely strong

20 Total exhaustion

Table 2. Local Perceived Pressure(LPP)

0 Nothing at all ("Nothing" means you don't feel any pressure)

1 Very slight

2 Slight

3 Medium (represents some but not very difficult)

4 Slightly serious

5 Serious ("Severe" is very difficult and tiring, but not very difficult to proceed. This level is about half of the "Maximum")

6 Between 5-7

7 Very serious ("very serious" you can continue, but you have to force yourself and you are very tired.)

8 between 7-9

9 Very very serious (almost to the maximum)

10 Maximum value ("Extremely Intense-Maximum Value" is an extremely intense level, for most people this is the most intense level they have experienced in their previous lives)

Table 3. System Usability Scale(SUS)

questions Strongly disagree Basically disagree Neutral Basically agree Strongly agree

1. I would be willing to use this exoskeleton frequently. 

2. I found that this exoskeleton does not need to be so complicated. 

3. I think the exoskeleton is easy to use. 

4. I think I need the help of a technician to use the exoskeleton. 

5. I find that the different functions in this exoskeleton are well integrated. 

6. I think this exoskeleton has too many uncoordinated content. 

7. I think most people will learn to use this exoskeleton soon. 

8. I find this exoskeleton is very clumsy and troublesome to use. 

9. I feel confident about using this exoskeleton. 

10. Before I can use the exoskeleton independently , I need to learn a lot. 

Comment 5:

PLOS requires an ORCID iD for the corresponding author in Editorial Manager on papers submitted after December 6th, 2016. Please ensure that you have an ORCID iD and that it is validated in Editorial Manager. To do this, go to ‘Update my Information’ (in the upper left-hand corner of the main menu), and click on the Fetch/Validate link next to the ORCID field. This will take you to the ORCID site and allow you to create a new iD or authenticate a pre-existing iD in Editorial Manager. Please see the following video for instructions on linking an ORCID iD to your Editorial Manager account: https://www.youtube.com/watch?v=_xcclfuvtxQ

Response to Comment 5:

 Thanks for your comments, the corresponding author have completed the ORCID linking according to the instruction. The corresponding author ORCID is 0000-0001-8661-0093.

Comments to the Author

Response to Reviewer #1:

Reviewer #1: Interesting study unfortunately with a small sample size and reasonable time of workload. The results illustrate a good potential of the passive exoskeleton to reduce muscle activity of the low back and upper arms, but users perceived more pressure on the shoulders, thighs, wrists and waist. Only 50 % of the users rated the usability of the equipment as acceptable. 

Comment 1:

So the questions is whether acceptance is farther reduced by wearing the exoskeleton for a longer time?

Response to Comment 1:

Thanks for your comments. Most of the subjects who scored below the acceptable standard expressed a neutral attitude towards whether they would use exoskeleton frequently. Huysamen et al. stated that the contact pressure caused by the exoskeleton is expected to increase over a longer period of use, and the discomfort will increase (Huysamen et al. 2018). Therefore, a longer period of use may lead to a decline in acceptance. We have made changes in the discussion section. In addition, an optimization plan based on feedback from subjects is being implemented, which may increase user acceptance in the future.

Comment 2:

Validity of the studie respectively to the benefit for the users might be better by a bigger sample size.

Response to Comment 2:

Thank you for your kind suggestion. We referred to several related articles when setting the sample size, and the size they select was between 6-10(Abdoli et al. 2006, Wehner et al. 2009, Graham et al. 2009). Now we also find that some researchers chose a larger sample size (18 and above) for experimental testing (Bosch et al. 2016, Ulrey et al 2013). As you said, more sample size can improve the validity of the study. We have added a statement that the experiment is based on a small sample size in the discussion section. We are trying to improve our experimental conditions. In the future, we will consider doubling the sample size to increase the reliability of experimental data and statistical test results.

References

Abdoli-E, M., Agnew, M. J., & Stevenson, J. M. (2006). An on-body personal lift augmentation device (PLAD) reduces EMG amplitude of erector spinae during lifting tasks. Clinical Biomechanics, 21(5), 456–465.

Bosch, T., Eck, J. van, Knitel, K., & Looze, M. de. (2016). The effects of a passive exoskeleton on muscle activity, discomfort and endurance time in forward bending work. Applied Ergonomics, 54, 212–217.

Graham, R. B., Agnew, M. J., & Stevenson, J. M. (2009). Effectiveness of an on-body lifting aid at reducing low back physical demands during an automotive assembly task: Assessment of EMG response and user acceptability. Applied Ergonomics, 40(5), 936–942.

Huysamen, K., Looze, M. de, Bosch, T., Ortiz, J., Toxiri, S., & O’Sullivan, L. W. (2018). Assessment of an active industrial exoskeleton to aid dynamic lifting and lowering manual handling tasks. Applied Ergonomics, 68, 125–131.

Ulrey, B. L., & Fathallah, F. A. (2013). Subject-specific, whole-body models of the stooped posture with a personal weight transfer device. Journal of Electromyography and Kinesiology, 23(1), 206–215.

Wehner, M., Rempel, D., & Kazerooni, H. (2009). Lower Extremity Exoskeleton Reduces Back Forces in Lifting. In ASME 2009 Dynamic Systems and Control Conference, Volume 2 (pp. 49–56).

Response to Reviewer #2:

Reviewer #2: The authors present an evaluation study for a developed passive exoskeleton. The following remarks might be useful to consider when finalizing the paper.

Introduction section

Comment 1:

- lower back pain should be linked to a musculoskeletal disorder rather than injury

Response to Comment 1:

 Thank you for the advice comment. ‘musculoskeletal injury’ has been changed to ‘musculoskeletal disorder’ in the introduction section.

Comment 2:

- LPP is named without introduction of the meaning

Response to Comment 2:

 Sorry for the confusion, LPP is a writing error. LPP has changed to LBP.

Comment 3:

- sources are quite old (e.g. LBP data from 2008) and should be updated

Response to Comment 3:

 Thank you for your reminder. We have updated the sources mentioned in the introduction section, including:

‘ of which low back pain (LBP) is the number one cause of disability in the world(3)’;

‘the indirect costs caused by LBP represented overall 0.68% of Spanish Gross Domestic Product(4)’;

‘ From 1990 to 2016, 12.8 million individuals with LBP had increased in China(5)’;

‘ LBP is among the biggest causes of absence from work(6).’;

‘ lifting aids are often not used due to their constraints(9)’. 

References

3. Hoy D, March L, Brooks P, Blyth F, Woolf A, Bain C, et al. The global burden of low back pain: estimates from the Global Burden of Disease 2010 study. annals of the rheumatic diseases. 2014;73(6):968-974.

4. Alonso-García M, Sarría-Santamera A. The Economic and Social Burden of Low Back Pain in Spain: A National Assessment of the Economic and Social Impact of Low Back Pain in Spain. spine. 2020;45(16).

5. Wu A, Dong W, Liu S, Cheung JPY, Kwan KYH, Zeng X, et al. The prevalence and years lived with disability caused by low back pain in China, 1990 to 2016: findings from the global burden of disease study 2016. pain. 2019;160(1):237-245.

6. Pistolesi F, Lazzerini B. Assessing the Risk of Low Back Pain and Injury via Inertial and Barometric Sensors. ieee transactions on industrial informatics. 2020;16(11):7199-7208.

9. Baltrusch SJ, Houdijk H, Dieën JHv, Bennekom CAMv, Kruif AJTCMd. Perspectives of End Users on the Potential Use of Trunk Exoskeletons for People With Low-Back Pain: A Focus Group Study. human factors. 2020;62(3):365-376.

Comment 4:

- there are studies published with subjective evaluation of exoskeletons (e.g. from Graham et al. 2009 and Dewi et al. 2018). This should be corrected.

Response to Comment 4:

 Thanks for your comments, we have deleted ‘and ignored the subjects' subjective evaluation of the exoskeleton’ in the manuscript.

Method section

Comment 5:

- it is not clear, whether and how the upper arms are supported by the exoskeleton

Response to Comment 5:

 Thanks for your comments. The assistance of the exoskeleton to the upper arms is realized with the help of the hooks, shoulder straps, back support and elastic units in Fig 1 of the manuscript. The subject first bends over to fix the hook with the box, as shown in Pic 1 a). When restoring upright, part of the box weight is transferred to the shoulders and back support by the straps, and the elastic unit releases the potential energy to provide assistance. The palms are located on both sides for auxiliary fixation. After returning to an upright position, the box weight is partly transferred to the shoulders as shown in Fig 1 b). This is one of the reasons for the higher LLP score of the shoulders. We have added how the upper arms are supported by the exoskeleton in the Method section.

a) bending over b) returning upright

Pic 1. Lifting assistance with exoskeleton

Comment 6:

- the EMG system seem to be wired - have there been any contacts with the exoskeleton? - this might have caused interferences

Response to Comment 6:

Dear reviewer, the subject wearing all the equipment for the tests was shown in the Pic 2 below. The lifting process were basically completed in the sagittal plane. The wire between the electrodes and the EMG system was located in the coronal plane, which would not restrict and interfere with the entire lifting process. It is worth mentioning that during the whole experiment, a laboratory staff member was arranged to master the position of the EMG system to avoid the displacement of the electrodes position or the discomfort of the subject caused by the fixed length of the wires.

Pic 2. wearing all the equipment for the tests

Testing procedures section

Comment 7:

- why did the subjects rate LPP only after the single lifting tasks?

Response to Comment 7:

 Thanks for your questions. The purpose of LPP scoring is to visually evaluate the perceived musculoskeletal pressure in contact with the exoskeleton, which is the most direct and obvious feeling after the subject has completed a single carrying task. Before the start of the formal experiment, we invited 3 laboratory researchers to wear exoskeleton for LPP scoring. The results showed that after the 15-minute repeated lifting task, the subjects were more willing to describe the current fatigue degree of each part, but could not accurately judge the current perceived pressure. After the single lifting task, they could clearly describe the difference of perceived pressure in different areas. In addition, Kirsten et al. also rated LPP after completing just five cyclical lifting and lowering (Huysamen et al. 2018). Therefore, we only required subjects to score LPP after completing the single lifting tasks in the formal experiment. 

Comment 8:

- it should be explained why the single lifting tasks were performed in addition to the repetitive lifting tasks

Response to Comment 8:

 Thanks for your questions. The experimental results of the 15-minute repetitive handling task can directly reflect the differences of EMG, oxygen consumption and subjective perception of fatigue with/without the exoskeleton. But we are also interested in whether there is a difference in the assisting effect between the single lifting tasks and repeated lifting tasks. The EMG results confirmed the existence of the difference. The LPP result after the single lifting tasks is also meaningful for discussion. In addition, some articles involving exoskeleton evaluation have designed the similar single lifting task modes(Bosch et al. 2016, Frost et al. 2009, Wehner et al. 2009, etc.). Therefore, we designed two task modes for subjects.

Comment 9:

- Figure 3 shows very un-ergonomic lifting, why was this way of lifting chosen?

Response to Comment 9:

 Thanks for your comments. Before the tests started, we asked each subject to complete the lifting tasks according to their personal lifting habits, and to maintain the consistency of lifting actions as much as possible. The training aims to simulate the scene of the workers' daily lifting process. Workers were not required to train in accordance with the most ergonomic lifting. After getting your comments, we paid a return visit to the workers who participated in the experiment and found that they generally developed the habit of un-ergonomic lifting in their daily work last week. We specially reminded and guided them. Thanks for your comments.

References

Huysamen, K., Looze, M. de, Bosch, T., Ortiz, J., Toxiri, S., & O’Sullivan, L. W. (2018). Assessment of an active industrial exoskeleton to aid dynamic lifting and lowering manual handling tasks. Applied Ergonomics, 68, 125–131.

Bosch, T., Eck, J. van, Knitel, K., & Looze, M. de. (2016). The effects of a passive exoskeleton on muscle activity, discomfort and endurance time in forward bending work. Applied Ergonomics, 54, 212–217.

Wehner, M., Rempel, D., & Kazerooni, H. (2009). Lower Extremity Exoskeleton Reduces Back Forces in Lifting. In ASME 2009 Dynamic Systems and Control Conference, Volume 2 (pp. 49–56).

Frost, D. M., Abdoli-E, M., & Stevenson, J. M. (2009). PLAD (personal lift assistive device) stiffness affects the lumbar flexion/extension moment and the posterior chain EMG during symmetrical lifting tasks. Journal of Electromyography and Kinesiology, 19(6).

Results section

Comment 10:

- Only half of the users rate the system usable...

Response to Comment 10:

We evaluated the usability of all 8 subjects who participated in the tests, and the results showed that only 4 subjects rated scores higher than 70. We have added ‘All subjects rated the system usability scores’ in the Usability section.

Comment 11:

- there are only effects for muscle activity not for perceived exertion or oxygen consumption - what is a possible explanation for this?

Response to Comment 11:

 Dear reviewer, the design of IPAE has not added external power source as a passive exoskeleton. The mechanical structure might achieve the boosting effect on local muscles and we thought subjects would experience the same energy demand regardless of whether they were wearing the exoskeleton before the test. The result of oxygen consumption shows IPAE cannot cause significant differences. This conforms to our hypothesis and is basically consistent with the conclusion drawn by Whitfield et al. (Whitfield et al. 2018). The insignificant difference in the perceived exertion may be related to the experimental sample size and task intensity. The test was based on a not big sample size. 50% of the subjects indicated that the task intensity was not high and it had not yet reached much significant fatigue after the tests. In addition, Rashedi et al. calibrated the perceived exertion with subjects before the test(Rashedi et al. 2014). This would improve the reliability in the next experiment.

References

Rashedi, E., Kim, S., Nussbaum, M. A., & Agnew, M. J. (2014). Ergonomic evaluation of a wearable assistive device for overhead work. Ergonomics, 57(12), 1864–1874.

Whitfield, B. H., Costigan, P. A., Stevenson, J. M., & Smallman, C. L. (2014). Effect of an on-body ergonomic aid on oxygen consumption during a repetitive lifting task. International Journal of Industrial Ergonomics, 44(1), 39–44.

Discussion Section

Comment 12:

- Reducing muscular activity does not reduce risk of injury per se

Response to Comment 12:

Thanks for your comments. We have deleted ‘thereby reducing the risk of muscle injury’ in the Discussion section.

Comment 13:

- It should be discussed what possible long term effects could be

Response to Comment 13:

Thanks for your suggestions. Most of the subjects who scored below the acceptable standard expressed a neutral attitude towards whether they would use exoskeleton frequently. Huysamen et al. stated that the contact pressure caused by the exoskeleton is expected to increase over a longer period of use, and the discomfort will increase (Huysamen et al. 2018). Therefore, a longer period of use may lead to a decline in acceptance as reviewer #1 mentioned . We have added ‘It can be speculated that longer use may reduce the acceptance of the exoskeleton’ in the Discussion section.

Comment 14:

- It is stated that age could have an effect on the SUS scores - with 27 years average and 4 years SD?

Response to Comment 14:

Thanks for your questions. When we analyzed the experimental results, it can be found that the SUS scores of the 4 older subjects (30.5±3.1 years, 63.8±4.8 scores) were significantly lower than other 4 younger subjects (24.3±1.7 years, 71.9±4.7scores). This finding has been added to the Discussion section. The specific scores of the SUS also indicate that younger subjects generally believed that IPAE was easy to learn, and they had a negative attitude to learning many things before using the IPAE. Therefore, it can be speculated that ‘the score of SUS may also be affected by the age of the subjects in this work’. This may be an interesting finding. In the further work, we will try to select two subjects of different ages for comparative experiments.

Comment 15:

- there are studies showing a reduction in energy consumption (e.g. Baltrusch et al. 2018 with LAEVO), this should be taken into account

Response to Comment 15:

Thanks for your suggestions. We have read the articles by Baltrusch et al. and added ‘But the studies by Baltrusch et al. has showed a reduction in energy consumption during lifting with the LAEVO’ (Baltrusch et al. 2018).

Comment 16:

- the authors do not discuss that other structures of the body might have a higher load due to the use of the exoskeleton

Response to Comment 16:

 Thanks for your comments. Before the formal tests, three laboratory researchers who voluntarily participated in the test evaluated the contact pressure of various parts of the body: the chest, abdomen, arms and upper back did not get obvious contact pressure. But there was significant contact pressure on the shoulders, waist, wrists and thighs. Therefore, we focused on whether the four parts of the body would have a high load based on previous feedback during the formal tests. All the subjects did not report that there were obvious pressures on other parts caused by the exoskeleton. We have added ‘All the subjects did not report that there was obvious stress caused by the exoskeleton in other structures of the body’ in the discussion section.

References

Baltrusch, S. J., Dieën, J. H. van, Bruijn, S. M., Koopman, A. S., Bennekom, C. A. M. van, & Houdijk, H. (2018). The Effect of a Passive Trunk Exoskeleton on Functional Performance and Metabolic Costs. International Symposium on Wearable Robotics, 22, 229–233.

Huysamen, K., Looze, M. de, Bosch, T., Ortiz, J., Toxiri, S., & O’Sullivan, L. W. (2018). Assessment of an active industrial exoskeleton to aid dynamic lifting and lowering manual handling tasks. Applied Ergonomics, 68, 125–131.

Conclusion Section

Comment 17:

- The study reveals low effects and high discomfort at the same time. This should be mentioned accordingly.

Response to Comment 17:

Thanks for your suggestions. We have modified the conclusion ‘the IPAE significantly reduced the muscle fatigue of both the low back and upper arms of subjects during lifting works, but the test results reveals low effects and high discomfort at the same time as a passive exoskeleton.’

Comment 18:

- In total the results currently do not support fully the drawn conclusions (e.g. reduction of muscular injury).

Response to Comment 18:

 Thanks for your comments. We have deleted the not rigorous inferences ‘ Therefore, it can effectively reduce the risk of workers' muscle injury.’

---

## [Decision Letter · Decision Letter 1]

5 Jan 2021

Effects of an industrial passive assistive exoskeleton on muscle activity, oxygen consumption and subjective responses during lifting tasks

PONE-D-20-26515R1

Dear Dr. Qu,

We’re pleased to inform you that your manuscript has been judged scientifically suitable for publication and will be formally accepted for publication once it meets all outstanding technical requirements.

Kind regards,

Peter Schwenkreis

Academic Editor

PLOS ONE

Additional Editor Comments (optional):

Reviewers' comments:

Reviewer's Responses to Questions

**Comments to the Author**

1. If the authors have adequately addressed your comments raised in a previous round of review and you feel that this manuscript is now acceptable for publication, you may indicate that here to bypass the “Comments to the Author” section, enter your conflict of interest statement in the “Confidential to Editor” section, and submit your "Accept" recommendation.

Reviewer #1: All comments have been addressed

Reviewer #2: All comments have been addressed

2. Is the manuscript technically sound, and do the data support the conclusions?

Reviewer #1: Yes

Reviewer #2: Yes

3. Has the statistical analysis been performed appropriately and rigorously? 

Reviewer #1: Yes

Reviewer #2: Yes

4. Have the authors made all data underlying the findings in their manuscript fully available?

Reviewer #1: Yes

Reviewer #2: Yes

5. Is the manuscript presented in an intelligible fashion and written in standard English?

Reviewer #1: Yes

Reviewer #2: Yes

6. Review Comments to the Author

Reviewer #1: The authors took account of reviewers comments and reworked their manuscript adequate. One mistake I found in Ethics Statement. Here the authors have to correct the word LPP into LBP that meens low back pain:....with no history of muscle injury and LBP in the past three months

Reviewer #2: Dear authors, thanks a lot for the additional work on your paper and for giving explanations to the questions that arised during the review.

7. PLOS authors have the option to publish the peer review history of their article (what does this mean?). If published, this will include your full peer review and any attached files.

Reviewer #1: No

Reviewer #2: No

---

## [Editor Report · Acceptance letter]

7 Jan 2021

PONE-D-20-26515R1 

Effects of an industrial passive assistive exoskeleton on muscle activity, oxygen consumption and subjective responses during lifting tasks 

Dear Dr. Qu:

I'm pleased to inform you that your manuscript has been deemed suitable for publication in PLOS ONE. Congratulations! Your manuscript is now with our production department. 

Kind regards, 

on behalf of

Dr. Peter Schwenkreis 

Academic Editor

PLOS ONE